# Hydrogen Sulfide Modulation of Matrix Metalloproteinases and CD147/EMMPRIN: Mechanistic Pathways and Impact on Atherosclerosis Progression

**DOI:** 10.3390/biomedicines12091951

**Published:** 2024-08-26

**Authors:** Constantin Munteanu, Anca Irina Galaction, Mădălina Poștaru, Mariana Rotariu, Marius Turnea, Corneliu Dan Blendea

**Affiliations:** 1Department of Biomedical Sciences, Faculty of Medical Bioengineering, University of Medicine and Pharmacy “Grigore T. Popa”, 700115 Iasi, Romania; anca.galaction@umfiasi.ro (A.I.G.); mariana.rotariu@umfiasi.ro (M.R.); marius.turnea@umfiasi.ro (M.T.); 2Neuromuscular Rehabilitation Clinic Division, Clinical Emergency Hospital “Bagdasar-Arseni”, 041915 Bucharest, Romania; 3Department of Medical-Clinical Disciplines, General Surgery, Faculty of Medicine, “Titu Maiorescu” University of Bucharest, 0400511 Bucharest, Romania; dan.blendea@prof.utm.ro

**Keywords:** hydrogen sulfide (H_2_S), matrix metalloproteinases (MMPs), CD147/EMMPRIN, atherosclerosis, inflammation, vascular remodeling

## Abstract

Atherosclerosis is a chronic inflammatory condition marked by endothelial dysfunction, lipid accumulation, inflammatory cell infiltration, and extracellular matrix (ECM) remodeling within arterial walls, leading to plaque formation and potential cardiovascular events. Key players in ECM remodeling and inflammation are matrix metalloproteinases (MMPs) and CD147/EMMPRIN, a cell surface glycoprotein expressed on endothelial cells, vascular smooth muscle cells (VSMCs), and immune cells, that regulates MMP activity. Hydrogen sulfide (H₂S), a gaseous signaling molecule, has emerged as a significant modulator of these processes including oxidative stress mitigation, inflammation reduction, and vascular remodeling. This systematic review investigates the mechanistic pathways through which H₂S influences MMPs and CD147/EMMPRIN and assesses its impact on atherosclerosis progression. A comprehensive literature search was conducted across PubMed, Scopus, and Web of Science databases, focusing on studies examining H₂S modulation of MMPs and CD147/EMMPRIN in atherosclerosis contexts. Findings indicate that H₂S modulates MMP expression and activity through transcriptional regulation and post-translational modifications, including S-sulfhydration. By mitigating oxidative stress, H₂S reduces MMP activation, contributing to plaque stability and vascular remodeling. H₂S also downregulates CD147/EMMPRIN expression via transcriptional pathways, diminishing inflammatory responses and vascular cellular proliferation within plaques. The dual regulatory role of H₂S in inhibiting MMP activity and downregulating CD147 suggests its potential as a therapeutic agent in stabilizing atherosclerotic plaques and mitigating inflammation. Further research is warranted to elucidate the precise molecular mechanisms and to explore H₂S-based therapies for clinical application in atherosclerosis.

## 1. Introduction

Atherosclerosis is a chronic inflammatory disease that primarily affects large and medium-sized arteries. It is characterized by the accumulation of lipids, inflammatory cells, and fibrous elements within the arterial wall, leading to the formation of atherosclerotic plaques [1,2,3,4,5]. The disease progresses through a series of stages, beginning with endothelial dysfunction, then plaque formation, and culminating in vascular remodeling. Understanding these stages is crucial for developing targeted therapies to mitigate the progression and consequences of atherosclerosis [6,7].

Endothelial dysfunction is one of the earliest events in the pathogenesis of atherosclerosis [8]. The endothelium, a monolayer of cells lining the blood vessels, is crucial in maintaining vascular homeostasis by regulating blood flow, coagulation, and inflammation [9]. Factors such as hypertension, hyperlipidemia, smoking, and diabetes can damage the endothelium, reducing its ability to produce nitric oxide (NO), a potent vasodilator and anti-inflammatory molecule. This dysfunction results in increased permeability of the endothelium, allowing low-density lipoprotein (LDL) cholesterol to infiltrate the arterial wall [10]. The oxidized form of LDL (ox-LDL) is particularly atherogenic and triggers a cascade of inflammatory responses [11]. Plaque formation begins with the retention and oxidation of LDL in the subendothelial space, leading to the recruitment of monocytes from the bloodstream [12]. These monocytes differentiate into macrophages, which engulf ox-LDL to become foam cells [13]. 

Foam cells are a hallmark of early atherosclerotic lesions known as fatty streaks. As the disease progresses, the accumulation of foam cells, T-lymphocytes, and smooth muscle cells (SMCs) contributes to plaque growth. SMCs start to migrate from the media to the intima, the place where they proliferate and secrete extracellular matrix components, including collagen and elastin, contributing to the fibrous cap of the plaque [13]. The fibrous cap stabilizes the plaque but also encroaches on the lumen of the artery, reducing blood flow [14]. The continued presence and activity of foam cells, along with the recruitment of smooth muscle cells (SMCs) from the media to the intima, initiate structural changes within the arterial wall.

Vascular remodeling refers to the structural changes that occur in the arterial wall in response to the growing plaque and the altered hemodynamic forces [15]. There are two main types of remodeling: outward (or positive) remodeling and inward (or negative) remodeling [16]. Outward remodeling involves the expansion of the arterial wall to preserve the lumen size, whereas inward remodeling results in luminal narrowing and reduced blood flow [17]. Advanced plaques often have a necrotic core composed of dead foam cells and extracellular lipids covered by a thin fibrous cap. The stability of this cap is crucial; its rupture can expose pro-thrombotic material to the bloodstream, resulting in the formation of a thrombus. Vascular remodeling alters the arterial wall’s structural integrity and stabilizes the atherosclerotic plaque. The continued inflammatory response within the plaque, driven by foam cells and SMCs, promotes further extracellular matrix degradation, ultimately weakening the fibrous cap and increasing the risk of plaque rupture and subsequent cardiovascular events [18].

Inflammation is a central feature of all stages of atherosclerosis. Various inflammatory cytokines and chemokines are involved in the recruitment and activation of immune cells within the plaque. Matrix metalloproteinases (MMPs) produced by macrophages and SMCs degrade extracellular matrix components, contributing to plaque instability. CD147/EMMPRIN, a regulator of MMPs, further modulates the inflammatory milieu and matrix degradation. Persistent inflammation and matrix degradation weaken the fibrous cap, increasing the risk of plaque rupture and subsequent thrombotic events [19].

Matrix metalloproteinases (MMPs) are a group of zinc-dependent endopeptidases involved in degrading extracellular matrix (ECM) components. Their importance in the context of atherosclerosis cannot be overstated [20], as they are involved in various stages of plaque development and stability. MMPs are responsible for breaking down various ECM components, such as collagen, elastin, gelatin, and proteoglycans. In the arterial wall, this activity is essential for tissue remodeling and repair. However, in the context of atherosclerosis, the dysregulated activity of MMPs contributes to the pathological remodeling of the vessel wall [21].

MMPs facilitate the migration and proliferation of vascular smooth muscle cells (VSMCs) from the media to the intima, which is crucial for developing the fibrous cap of atherosclerotic plaques. They also degrade the ECM within the intima, which allows for the expansion of the lipid core and further accumulation of inflammatory cells [22]. The balance between ECM synthesis and degradation heavily influences the stability of atherosclerotic plaques. MMPs, particularly MMP-2 and MMP-9, are involved in the degradation of collagen and other structural proteins in the fibrous cap of plaques [23]. A stable plaque typically has a thick fibrous cap and a small, well-contained lipid core [24]. Conversely, an unstable plaque has a thin fibrous cap and a large lipid core and is rich in inflammatory cells. The overexpression and activation of MMPs in atherosclerotic plaques lead to the thinning of the fibrous cap by degrading its collagen content, making it more susceptible to rupture. Plaque rupture is a critical event that can lead to acute cardiovascular events such as myocardial infarction and stroke, as it exposes the pro-thrombotic core to the bloodstream, triggering thrombosis [25].

MMPs are not only involved in ECM degradation but also play a role in modulating inflammatory responses within atherosclerotic plaques. MMPs can activate pro-inflammatory cytokines and chemokines, which further recruit and activate inflammatory cells such as macrophages and T-lymphocytes. These cells, in turn, produce more MMPs, creating a feedback loop that exacerbates inflammation and plaque instability [26]. Additionally, MMPs can degrade membrane-bound cytokine precursors, releasing active cytokines that perpetuate the inflammatory cycle. Given their central role in plaque stability and atherosclerosis progression, MMPs present a potential therapeutic target. Inhibiting MMP activity could stabilize plaques by preserving the integrity of the fibrous cap and reducing inflammation. However, broad-spectrum MMP inhibitors have shown limited success in clinical trials due to adverse side effects and lack of specificity. Thus, there is ongoing research to develop more selective MMP inhibitors that can effectively target the pathological activity of MMPs without disrupting their physiological functions [27].

CD147, also known as extracellular matrix metalloproteinase inducer (EMMPRIN), is a transmembrane glycoprotein integral to regulating matrix metalloproteinases (MMPs) [28]. It is broadly expressed in various tissues, including endothelial cells, vascular smooth muscle cells (VSMCs), and immune cells [29]. CD147 is pivotal in physiological and pathological processes, particularly in inflammation and vascular remodeling, both of which are critical in the progression of atherosclerosis [30].

CD147 regulates MMP expression and activity through direct interactions, intracellular signaling pathways, and glycosylation. It induces the production of MMPs like MMP-1, MMP-2, MMP-3, MMP-9, and MMP-14 by interacting directly with these enzymes, activating signaling pathways such as MAPK/ERK, and through its glycosylation status. Highly glycosylated forms of CD147 are more effective in stimulating MMP expression. By modulating MMP activity, CD147 influences the extracellular matrix’s (ECM) degradation, which is essential for tissue remodeling and cell migration. While this process is crucial for normal physiological functions such as wound healing, it can become detrimental in pathological conditions like atherosclerosis [31].

In inflammation, CD147 plays a significant role by recruiting and activating various immune cells, including monocytes, macrophages, and T lymphocytes [32]. It promotes the adhesion and migration of monocytes into the vascular intima, where they differentiate into macrophages and contribute to the inflammatory milieu of atherosclerotic plaques [33]. CD147 also enhances the production of pro-inflammatory cytokines like TNF-α, IL-6, and IL-1β, amplifying the inflammatory response and stimulating more MMP production. Furthermore, its interaction with cyclophilins, particularly cyclophilin A, enhances its pro-inflammatory and MMP-inducing effects, thereby perpetuating the cycle of inflammation and ECM degradation [34].

In vascular remodeling, CD147 facilitates structural changes in the vessel wall during atherosclerosis. It promotes the migration of VSMCs from the media to the intima and after their proliferation, which contributes to the formation of the fibrous cap and vessel wall thickening, which are essential for plaque growth and stability [35]. By regulating MMP activity, CD147 controls ECM degradation, allowing vascular structure remodeling to accommodate the growing atherosclerotic plaque. Additionally, CD147 is involved in angiogenesis within the plaque, which, while providing nutrients to the plaque, also increases the risk of intraplaque hemorrhage and further inflammation [36].

Hydrogen sulfide (H_2_S) has emerged as a significant gasotransmitter alongside nitric oxide (NO) and carbon monoxide (CO), playing vital roles in numerous physiological and pathological processes. H_2_S exhibits potent antioxidant properties by directly scavenging reactive oxygen species (ROS) and upregulating endogenous antioxidant defenses. It enhances the activity of key antioxidant enzymes such as superoxide dismutase (SOD), glutathione peroxidase (GPx), and catalase, which collectively mitigate oxidative stress by neutralizing ROS and preventing oxidative damage to cellular components [37]. This antioxidant capacity is particularly beneficial in vascular health, where oxidative stress plays a crucial role in endothelial dysfunction, inflammation, and the progression of atherosclerosis [38].

Additionally, H_2_S acts as a vasodilator, contributing to regulating blood pressure and vascular tone [39]. It induces vasodilation primarily by activating ATP-sensitive potassium (K_ATP) channels in vascular smooth muscle cells (VSMCs) [40]. H_2_S also modulates the nitric oxide (NO) signaling pathway by enhancing the bioavailability of NO, another critical vasodilator [41]. Furthermore, H_2_S has been shown to inhibit the proliferation and migration of VSMCs, processes that are central to the development of vascular pathologies such as atherosclerosis. By curbing VSMC proliferation, H_2_S helps prevent the thickening of the vessel wall and the formation of neointimal lesions, thus protecting against vascular occlusion and remodeling [42]. Beyond its vasodilatory and antioxidant roles, H_2_S also exhibits anti-inflammatory effects [43], which are critical in mitigating vascular inflammation and atherosclerosis [44]. 

H_2_S’s multifaceted roles in regulating oxidative stress, vascular tone, cellular proliferation, and inflammation underscore H_2_S‘s therapeutic potential in treating cardiovascular diseases. The development of H_2_S-releasing compounds and their application in clinical settings represents a promising avenue for preventing and managing conditions characterized by endothelial dysfunction, oxidative stress, and inflammation, such as hypertension, atherosclerosis, and heart failure [45].

The objective of this review is threefold: to provide a comprehensive understanding of hydrogen sulfide (H_2_S) modulation of matrix metalloproteinases (MMPs) and CD147/EMMPRIN, to elucidate the mechanistic pathways through which H_2_S influences the progression of atherosclerosis, and to explore the therapeutic potential of targeting these pathways for the treatment of atherosclerosis. The mechanistic pathways through which H_2_S affects atherosclerosis involve its antioxidant, anti-inflammatory, and vasodilatory properties. The review aims to detail these pathways, highlighting the interplay between H_2_S, MMPs, and CD147 in the context of atherosclerosis.

In this review, we conducted a comprehensive literature search across PubMed, Scopus, and Web of Science using specific keywords such as “hydrogen sulfide”, “matrix metalloproteinases”, “CD147”, “EMMPRIN”, and “atherosclerosis”, and we applied Boolean operators to refine our search, targeting studies published from 2000 to the present. Inclusion criteria focused on original research articles, reviews, and meta-analyses that investigated the roles of H₂S, MMPs, and CD147, providing mechanistic insights and involving human, animal, or in vitro models of atherosclerosis. Exclusion criteria ruled out studies unrelated to cardiovascular diseases, lacking full-text access, or that were not peer-reviewed. Data extraction followed a structured approach, capturing critical information on study design, key findings, and the relevance of each study to understanding H₂S’s role in modulating MMP activity and CD147 expression. We assessed the quality and validity of the studies based on methodological rigor, reproducibility, and peer-review status. This approach ensured that only high-quality studies were included, enabling a comprehensive synthesis of the current knowledge on the therapeutic potential of H₂S in atherosclerosis.

## 2. Role of Hydrogen Sulfide in Atherosclerosis

Hydrogen sulfide (H_2_S), while historically recognized as a toxic gas, is an endogenously produced gasotransmitter that is recognized for its significant role in various physiological and pathological processes, including atherosclerosis. It is synthesized primarily through the enzymatic activities of cystathionine γ-lyase (CSE), cystathionine β-synthase (CBS), and 3-mercaptopyruvate sulfurtransferase (3-MST) [37]. These three enzymatic pathways are tightly regulated to ensure proper H_2_S levels in the body. The balance of H_2_S production and degradation is crucial for maintaining its physiological functions, which include modulating inflammation, vasodilation, and antioxidant defense. Disruption in H_2_S production has been associated with various diseases, including cardiovascular diseases, neurodegenerative disorders, and metabolic syndromes [46].

Cystathionine γ-lyase (CSE), predominantly found in the cardiovascular system, liver, kidney, and brain, catalyzes the conversion of cystathionine to cysteine, α-ketobutyrate, and ammonia, leading to the production of H_2_S. The activity of CSE is crucial in the cardiovascular system, where it helps regulate blood pressure and protect against atherosclerosis. The expression of CSE can be influenced by various factors, including dietary intake of sulfur-containing amino acids and hormonal regulation, highlighting its dynamic role in H_2_S production [47]. 

Cystathionine β-synthase (CBS) is primarily expressed in the central nervous system and the liver. CBS catalyzes the condensation of serine and homocysteine to form cystathionine, which is further metabolized by CSE to produce H_2_S. The activity of CBS is regulated by the availability of its substrates and cofactors, such as pyridoxal phosphate (vitamin B6) and S-adenosylmethionine (SAM), as well as by various signaling molecules [48]. 

3-Mercaptopyruvate sulfurtransferase (3-MST) is an enzyme found in the mitochondria and cytosol of many tissues, including the brain, liver, and kidneys. It catalyzes the transfer of a sulfur atom from 3-mercaptopyruvate, which is produced from cysteine, by cysteine aminotransferase (CAT), to a thiol acceptor, producing H_2_S [49]. 

Hydrogen sulfide (H₂S) serves a dual role in the body, acting as both a protective signaling molecule at physiological levels and a potential toxin when levels are imbalanced. Maintaining an optimal balance of H₂S is crucial for health, as both excessive and insufficient concentrations can lead to pathological conditions. Elevated H₂S levels have been associated with detrimental effects such as hypotension and neurological disturbances, while insufficient H₂S production is linked to impaired vascular function, increased oxidative stress, and chronic inflammation, all of which contribute to the progression of cardiovascular diseases like atherosclerosis. This delicate balance highlights the nuanced role of H₂S in physiological functions, where tight regulation is essential for preserving cardiovascular and systemic health [37].

One of the primary functions of H_2_S is vasodilation, which is critical for regulating blood pressure and ensuring adequate blood flow to tissues. H_2_S induces vasodilation primarily by activating ATP-sensitive potassium (K_ATP) channels in vascular smooth muscle cells (VSMCs). The activation of these channels leads to membrane hyperpolarization and a subsequent decrease in intracellular calcium levels, causing relaxation of the VSMCs and dilation of the blood vessels. Additionally, H_2_S interacts with the nitric oxide (NO) signaling pathway, enhancing the vasodilatory effects of NO. This synergistic relationship between H_2_S and NO involves the inhibition of phosphodiesterase (PDE), which increases cyclic guanosine monophosphate (cGMP) levels, further promoting vasodilation. This dual mechanism of action underscores the vital role of H_2_S in maintaining vascular tone and preventing hypertension, a risk factor for atherosclerosis [50].

H_2_S also exhibits potent antioxidant capabilities, which are crucial in preventing the oxidative damage associated with atherosclerosis. Oxidative stress, characterized by an imbalance between reactive oxygen species (ROS) production and antioxidant defenses, is a key factor in the pathogenesis of atherosclerosis. H_2_S directly scavenges ROS, reducing oxidative damage to endothelial cells and other vascular components. Additionally, H_2_S upregulates endogenous antioxidant defenses by enhancing the activity of enzymes such as superoxide dismutase (SOD), catalase, and glutathione peroxidase (GPx). By mitigating oxidative stress, H_2_S helps preserve endothelial function, which is crucial for preventing the initiation and progression of atherosclerotic lesions [51].

Inflammation is another critical process in the development of atherosclerosis, and H_2_S plays a significant role in modulating inflammatory responses. It inhibits the activation of nuclear factor kappa B (NF-κB), a key transcription factor involved in the inflammatory response. By reducing NF-κB activation, H_2_S decreases the expression of pro-inflammatory cytokines, adhesion molecules, and chemokines. This action diminishes the recruitment and activation of inflammatory cells, such as monocytes and macrophages, within the vascular wall. Furthermore, H_2_S promotes the production of anti-inflammatory cytokines and mediators, contributing to the resolution of inflammation and promoting tissue repair. These anti-inflammatory effects are vital in preventing the chronic inflammation that drives the progression of atherosclerosis [52,53]. H_2_S has been shown to influence cellular processes that are directly related to the stability of atherosclerotic plaques. It inhibits the proliferation and migration of VSMCs, which are key events in the formation of neointimal lesions and plaque development. By curbing these processes, H_2_S helps maintain the structural integrity of the arterial wall and prevents the excessive thickening that characterizes atherosclerotic plaques. H_2_S also plays a role in reducing vascular calcification, a common feature of advanced atherosclerosis, by inhibiting the osteogenic differentiation of VSMCs. By maintaining ECM integrity and reducing inflammation, H_2_S enhances the stability of atherosclerotic plaques, reducing the risk of rupture and subsequent cardiovascular events [43].

## 3. H_2_S Modulation of MMPs

Hydrogen sulfide (H_2_S) significantly influences matrix metalloproteinase (MMP) activity through direct inhibition and the regulation of oxidative stress. These interactions are crucial in the pathogenesis and progression of atherosclerosis, impacting plaque stability and vascular remodeling. One of the primary mechanisms by which H_2_S influences MMP activity is through direct inhibition of the proteolytic activity of MMPs. MMPs are a family of zinc-dependent proteases involved in degrading extracellular matrix (ECM) components, essential for tissue remodeling and repair. MMPs are synthesized as inactive zymogens (pro-MMPs) that require activation through proteolytic cleavage. H_2_S can interfere with this activation process by interacting with the catalytic sites of MMPs, particularly the zinc ions at the active site, thereby inhibiting their enzymatic activity. This direct inhibition prevents the excessive breakdown of ECM components, maintaining the structural integrity of the fibrous cap that covers atherosclerotic plaques. A stable fibrous cap is crucial for preventing plaque rupture, which can lead to thrombus formation and subsequent cardiovascular events [54].

However, excessive MMP activity can destabilize atherosclerotic plaques, making them prone to rupture and causing acute cardiovascular events. H_2_S interacts with the catalytic sites of MMPs, particularly the zinc ion in the active site, inhibiting their proteolytic activity. This direct inhibition helps maintain the structural integrity of the ECM, preventing excessive degradation that would otherwise contribute to plaque instability and rupture [55,56].

H_2_S also modulates MMP activity indirectly by regulating oxidative stress, a key factor in the activation and expression of MMPs. ROS, such as superoxide anions and hydrogen peroxide, can activate latent MMPs and increase their expression through redox-sensitive signaling pathways, including the nuclear factor kappa B (NF-κB) pathway. H_2_S exerts potent antioxidant effects by scavenging ROS and upregulating endogenous antioxidant defenses. By reducing ROS levels, H_2_S prevents the oxidative activation of MMPs, thereby inhibiting their proteolytic activity and subsequent ECM degradation [57].

Additionally, H_2_S influences MMP activity through the modulation of signaling pathways that regulate MMP expression [58]. For instance, H_2_S can inhibit the NF-κB pathway, a major regulator of inflammation and MMP expression. NF-κB activation leads to the transcription of various pro-inflammatory cytokines and MMPs, promoting vascular inflammation and ECM degradation. By inhibiting NF-κB activation, H_2_S reduces the transcription of MMP genes, thereby decreasing MMP levels and activity [59]. Furthermore, H_2_S can modulate other signaling pathways, such as the mitogen-activated protein kinase (MAPK) pathway, which is involved in cell proliferation, differentiation, and MMP expression [60]. Through these pathways, H_2_S exerts a broad regulatory effect on MMP activity, contributing to the stabilization of atherosclerotic plaques.

Emerging evidence suggests that H_2_S may also influence MMP activity through epigenetic mechanisms. Epigenetic modifications, such as DNA methylation and histone acetylation, can regulate gene expression without altering the DNA sequence. H_2_S has been shown to affect the activity of various epigenetic enzymes, potentially influencing the expression of MMPs. For example, H_2_S can inhibit histone deacetylases (HDACs), leading to changes in chromatin structure and gene expression. By modulating these epigenetic processes, H_2_S can indirectly regulate MMP expression and activity, contributing to its protective effects in atherosclerosis [61,62,63].

H_2_S impacts the expression and activity of MMPs in vascular cells and atherosclerotic plaques. Through its multifaceted mechanisms, H_2_S contributes to the stabilization of atherosclerotic plaques, reducing the risk of acute cardiovascular events such as myocardial infarction and stroke. H_2_S affects MMP expression at the transcriptional level in various vascular cells, including endothelial cells, vascular smooth muscle cells (VSMCs), and macrophages. One of the primary ways H_2_S modulates MMP expression is by influencing signaling pathways that regulate gene transcription. For example, H_2_S inhibits the NF-κB pathway, which is crucial for inflammation and MMP expression. NF-κB activation leads to increased transcription of MMP genes, such as MMP-9, which plays a significant role in ECM degradation within atherosclerotic plaques. By inhibiting NF-κB activation, H_2_S reduces the transcription of these MMP genes, leading to lower MMP levels in vascular cells and plaques [21].

In VSMCs [54], H_2_S inhibits the expression and activity of MMPs, which are involved in the migration and proliferation of these cells. The migration and proliferation of VSMCs from the media to the intima are key processes in developing atherosclerotic plaques and neointimal lesions. By reducing MMP activity, H_2_S prevents the degradation of the ECM, thereby inhibiting VSMC migration and proliferation. This action helps stabilize the plaque and reduce the risk of lumen occlusion [64,65].

Macrophages, which differentiate into foam cells upon ingesting oxidized low-density lipoprotein (ox-LDL), are significant sources of MMPs in atherosclerotic plaques. H_2_S modulates macrophage behavior, reducing the expression of pro-inflammatory cytokines and MMPs. By attenuating the inflammatory response and MMP production in macrophages, H_2_S reduces ECM degradation and plaque instability. Furthermore, H_2_S promotes the clearance of apoptotic cells by macrophages, a process known as efferocytosis, which is crucial for resolving inflammation and stabilizing plaques [66,67]. The ability of H_2_S to modulate these critical processes highlights its therapeutic potential in the management of atherosclerosis (Table 1). 

## 4. CD147/EMMPRIN and Its Regulation by H_2_S

CD147, also known as extracellular matrix metalloproteinase inducer (EMMPRIN), is a transmembrane glycoprotein prominently involved in the regulation of matrix metalloproteinases (MMPs). It plays a crucial role in various physiological and pathological processes, including inflammation, tissue remodeling, and tumor progression [71]. In the context of vascular biology, CD147 is essential for modulating MMP activity, which is critical for extracellular matrix (ECM) degradation and the pathogenesis of atherosclerosis [72]. CD147 induces several MMPs, including MMP-1, MMP-2, MMP-3, and MMP-9, through multiple mechanisms [73]. Firstly, CD147 directly interacts with MMPs and their proenzymes, enhancing their activation and activity. This interaction is crucial in tissues requiring rapid ECM remodeling. Secondly, CD147 activates intracellular signaling cascades, such as the mitogen-activated protein kinase (MAPK) and nuclear factor kappa B (NF-κB) pathways, which lead to the transcriptional upregulation of MMP genes, increasing MMP synthesis and secretion. Lastly, the glycosylation status of CD147 significantly affects its ability to induce MMP production, with highly glycosylated forms of CD147 being more effective in stimulating MMP expression and activity than low-glycosylated forms [72]. In Type 2 Diabetes Mellitus (T2DM), excessive glucose availability can lead to abnormal glycosylation patterns of CD147, potentially increasing its MMP-inducing activity. This abnormal glycosylation may exacerbate the inflammatory and degradative processes within the vascular walls, contributing to the accelerated progression of atherosclerosis observed in diabetic patients. Thus, the glycosylation of CD147 represents a plausible mechanistic link between glucotoxicity and vascular complications in T2DM, warranting further investigation into its role as a potential therapeutic target.

CD147 is widely expressed in various vascular cells, including endothelial cells, vascular smooth muscle cells (VSMCs), and immune cells such as macrophages and T lymphocytes. Its expression is upregulated in response to inflammatory stimuli and oxidative stress, conditions commonly associated with atherosclerosis [30]. In endothelial cells, CD147 expression increases during inflammation, contributing to increased permeability and leukocyte adhesion characteristic of endothelial dysfunction. By inducing MMPs, CD147 facilitates basement membrane degradation, allowing leukocyte transmigration and contributing to vascular inflammation [31]. In VSMCs, CD147 promotes MMP production, which is essential for cell migration and proliferation during atherosclerotic plaque formation. MMP activity induced by CD147 degrades ECM components, facilitating VSMC migration from the media to the intima, a key process in plaque development and vascular remodeling. In immune cells, CD147 regulates inflammatory responses and MMP production. High CD147 levels in macrophages and T lymphocytes within plaques enhance MMP activity, contributing to ECM degradation and plaque instability [35,72,74].

The upregulation of CD147 and its role in MMP induction have significant pathological implications in atherosclerosis. High levels of CD147 and MMPs are associated with increased plaque instability and a higher risk of rupture, leading to acute cardiovascular events such as myocardial infarction and stroke [72]. Targeting CD147 to reduce MMP activity and preserve ECM integrity is a potential therapeutic strategy for stabilizing atherosclerotic plaques. Inhibiting CD147 function could reduce MMP activity, preserve ECM integrity, and enhance plaque stability, thereby reducing the likelihood of rupture and subsequent cardiovascular complications [75]. CD147 interacts with various molecules, including integrins and cyclophilins, facilitating multiple processes essential for plaque development, progression, and stability. Integrins are transmembrane receptors that mediate cell adhesion to the ECM and play vital roles in cell migration, signaling, and survival. CD147 interacts with integrins to regulate these functions in the context of atherosclerosis. The interaction between CD147 and integrins enhances the adhesive properties of endothelial cells, VSMCs, and immune cells, which are crucial for recruiting and retaining inflammatory cells at sites of endothelial dysfunction. Integrins such as α3β1 and α6β1 associate with CD147, facilitating cell adhesion to the ECM and promoting VSMC migration, a critical step in forming the fibrous cap of atherosclerotic plaques. Furthermore, CD147–integrin interactions activate intracellular signaling pathways that regulate cell survival, proliferation, and differentiation. For example, engaging CD147 with integrins can activate the MAPK/ERK and PI3K/Akt pathways, promoting VSMC proliferation and survival. These processes contribute to vessel wall thickening and plaque progression [76].

Cyclophilins, proteins with peptidyl-prolyl isomerase activity involved in protein folding and trafficking, interact with CD147 to modulate inflammatory responses and MMP activity. The binding of cyclophilins to CD147 enhances the pro-inflammatory effects of both molecules. Cyclophilin A (CypA), secreted in response to oxidative stress and inflammation, binds to CD147 on immune cells, endothelial cells, and VSMCs. This interaction induces the secretion of pro-inflammatory cytokines such as TNF-α, IL-6, and MCP-1, exacerbating the inflammatory milieu within atherosclerotic plaques. The CD147–cyclophilin interaction also enhances MMP induction, particularly MMP-9. CypA binding to CD147 stimulates the NF-κB signaling pathway, increasing MMP transcription and activity. This upregulation facilitates ECM degradation, weakening the fibrous cap of plaques and increasing rupture risk [77].

Hydrogen sulfide (H_2_S) modulates CD147 expression and activity through several mechanistic pathways impacting cellular functions, inflammatory responses, and ECM remodeling. These pathways involve direct and indirect interactions regulating gene expression, protein activity, and signaling cascades, contributing to H_2_S’s protective effects in atherosclerosis and other vascular diseases. One of the primary mechanisms by which H_2_S influences CD147 is through the regulation of its gene expression. H_2_S can modulate transcription factors and signaling pathways that control CD147 transcription. H_2_S inhibits NF-κB activation, a key regulator of inflammation and gene expression. NF-κB activation leads to pro-inflammatory gene transcription, including CD147. By suppressing NF-κB, H_2_S reduces CD147 expression, decreasing MMP activity and ECM degradation, contributing to plaque stability. H_2_S also modulates the hypoxia-inducible factor 1-alpha (HIF-1α) pathway, which is involved in the cellular response to hypoxia. HIF-1α upregulates CD147 under hypoxic conditions commonly found in atherosclerotic plaques. By modulating HIF-1α, H_2_S can influence CD147 expression, potentially reducing hypoxia-induced inflammatory responses and MMP activity [70,78,79,80].

H_2_S influences CD147 activity through post-translational modifications, which can affect the stability, localization, and function of CD147. H_2_S can modify proteins via S-sulfhydration, adding a sulfur atom to cysteine residues. This modification can alter protein activity, stability, and interactions [68]. S-sulfhydration of CD147 may influence its interaction with other proteins, such as integrins and cyclophilins, thereby modulating its role in MMP induction and inflammatory signaling. The antioxidant properties of H_2_S play a significant role in modulating MMP activity and CD147 function. By scavenging reactive oxygen species (ROS) and enhancing the activity of endogenous antioxidant enzymes such as superoxide dismutase (SOD) and glutathione peroxidase (GPx), H_2_S reduces oxidative stress, which can activate MMPs through redox-sensitive pathways. By lowering ROS levels, H_2_S prevents the oxidative activation of MMPs, thereby reducing ECM degradation and plaque instability. Additionally, the antioxidant effects of H_2_S help maintain the stability and proper function of CD147, preventing its overexpression and excessive MMP induction [69,81,82,83].

H_2_S modulates CD147 activity through its anti-inflammatory effects, which are crucial in the context of atherosclerosis. H_2_S inhibits the production of pro-inflammatory cytokines such as TNF-α, IL-6, and IL-1β, which can upregulate CD147 expression. By reducing cytokine levels, H_2_S decreases CD147-mediated inflammatory responses and MMP activity, contributing to the stabilization of atherosclerotic plaques. Furthermore, H_2_S reduces the expression of adhesion molecules, such as ICAM-1 and VCAM-1, which are involved in leukocyte recruitment to the endothelium. By inhibiting leukocyte adhesion and migration, H_2_S decreases CD147-mediated inflammatory responses within the vascular wall. The impact of H_2_S on CD147-mediated MMP regulation and inflammatory responses highlights its therapeutic potential in atherosclerosis [35,72,74]. 

## 5. Potential Therapeutic Strategies Targeting H_2_S Pathways to Modulate MMP and CD147 Activity

Targeting hydrogen sulfide (H_2_S) pathways offers a promising therapeutic approach to modulate matrix metalloproteinase (MMP) and CD147/EMMPRIN activity, thereby addressing key aspects of atherosclerosis progression and plaque stability (Figure 1). Several strategies can be employed to harness the protective effects of H_2_S in cardiovascular health.

H_2_S donors are compounds designed to release H_2_S in a controlled manner, providing sustained therapeutic effects. Sodium hydrosulfide (NaHS) and sodium sulfide (Na2S) are inorganic salts that rapidly release H_2_S upon administration, offering immediate benefits in reducing MMP activity and inflammation. However, their rapid release can lead to short-lived effects and potential side effects, such as hypotension, respiratory distress, and neurological disturbances, necessitating careful dosing and monitoring. Overexpression or excessive administration of H₂S could disrupt cellular homeostasis, leading to oxidative stress or interfering with mitochondrial function. GYY4137 is a slow-releasing H_2_S donor that provides a more sustained release of H_2_S, mimicking the endogenous production of the gas. GYY4137 has shown promising results in preclinical studies, reducing oxidative stress, inflammation, and MMP activity, leading to improved plaque stability. AP39 and AP123 are mitochondria-targeted H_2_S donors that specifically deliver H_2_S to the mitochondria, enhancing cellular bioenergetics and reducing mitochondrial ROS production. These compounds have demonstrated the potential in reducing endothelial dysfunction and MMP activity in vascular cells [84].

Increasing the body’s natural production of H_2_S can be an effective therapeutic strategy. Upregulating the expression or activity of Cystathionine γ-lyase (CSE) and Cystathionine β-synthase (CBS), the primary enzymes involved in H_2_S synthesis, can boost endogenous H_2_S levels. Pharmacological activators of these enzymes are being investigated for their potential to enhance H_2_S production and exert cardiovascular protective effects [85]. Dietary interventions that increase the intake of sulfur-containing amino acids such as cysteine and methionine can also enhance H_2_S production. Foods rich in these amino acids, such as garlic, onions, and cruciferous vegetables, may provide cardiovascular benefits through increased H_2_S synthesis [86].

Combining H_2_S donors or enhancers with other cardiovascular therapies can provide synergistic benefits. Statins, widely used to lower cholesterol levels and reduce cardiovascular risk, can be combined with H_2_S donors to enhance the anti-inflammatory and antioxidant effects, further reducing MMP activity and stabilizing plaques. Angiotensin-converting enzyme (ACE) inhibitors, such as Zofenopril, which also releases H_2_S, can provide dual benefits of blood pressure reduction and enhanced H_2_S-mediated protective effects. This combination can improve endothelial function and reduce atherosclerosis progression. Additionally, co-administering H_2_S donors with antioxidants like vitamin C or E can amplify the reduction of oxidative stress, enhancing the overall protective effects on the vascular system [87,88].

Advanced genetic approaches can be used to modulate H_2_S pathways and their effects on MMPs and CD147. Gene therapy involves delivering genes that encode for H_2_S-producing enzymes (CSE, CBS) to enhance endogenous H_2_S production. Viral vectors or nanoparticle-based delivery systems can be used to target specific tissues, such as the vascular endothelium or smooth muscle cells. RNA interference (RNAi) techniques, using small interfering RNA (siRNA) or short hairpin RNA (shRNA), can knock down the expression of CD147, reducing its activity and subsequent MMP induction. Combining RNAi approaches with H_2_S donors can provide a comprehensive strategy to mitigate atherosclerosis [89,90].

Developing targeted delivery systems can enhance the efficacy and safety of H_2_S-based therapies. Encapsulating H_2_S donors in nanoparticles can provide targeted and controlled release of H_2_S, minimizing systemic side effects and enhancing local therapeutic effects on atherosclerotic plaques. Injectable hydrogels loaded with H_2_S donors can provide sustained release of H_2_S at specific sites of vascular injury or atherosclerotic plaques, improving plaque stability and reducing inflammation [91].

The development of H_2_S-releasing drugs has made significant progress in recent years, driven by the recognition of H_2_S’s protective effects on cardiovascular health. Several H_2_S donors and H_2_S-enhancing compounds are in various stages of preclinical and clinical development, showing potential to treat atherosclerosis and other cardiovascular diseases [92,93]. 

Identifying biomarkers for H_2_S activity, MMP expression, and CD147 levels can help monitor disease progression and treatment efficacy. Research should aim to develop non-invasive biomarkers that can be easily measured in clinical settings [94,95].

The therapeutic potential of H_2_S in treating atherosclerosis is promising, yet several gaps remain in our understanding that need to be addressed through future research. The development of H_2_S-releasing drugs, such as slow-releasing compounds like GYY4137 and mitochondria-targeted donors like AP39, has shown considerable promise in preclinical studies [5,84,96,97]. 

Despite these advancements, the long-term safety and efficacy of H_2_S-based therapies require thorough investigation. Long-term studies in animal models and subsequent clinical trials are necessary to evaluate potential side effects, the sustainability of therapeutic effects, and the impact on overall cardiovascular health. Understanding individual variability in response to H_2_S therapy is also crucial for developing personalized treatment approaches [98,99,100]. 

Moreover, advanced drug delivery systems, such as nanoparticles and hydrogels, should be developed to provide targeted and sustained release of H_2_S at specific sites of vascular injury or atherosclerotic plaques [101,102]. These systems can enhance the therapeutic index of H_2_S-based treatments by ensuring localized and prolonged H_2_S release. Finally, future research should aim to elucidate the detailed molecular mechanisms through which H_2_S modulates MMPs and CD147/EMMPRIN, as well as the context-dependent effects of H_2_S in different stages of atherosclerosis [103]. Addressing these research gaps will be critical for translating the protective effects of H_2_S into effective clinical therapies for atherosclerosis and improving cardiovascular outcomes.

## 6. Conclusions

Hydrogen sulfide (H_2_S) has emerged as a crucial modulator of cardiovascular health, particularly in the context of atherosclerosis. Its protective role is underscored by its ability to enhance endothelial function, stabilize atherosclerotic plaques, and modulate key molecular pathways involved in the disease’s progression. H₂S stabilizes atherosclerotic plaques by inhibiting the activity of matrix metalloproteinases (MMPs), which are critical enzymes involved in the degradation of the extracellular matrix (ECM). MMPs, particularly MMP-2 and MMP-9, degrade collagen and other fibrous cap structural components that cover atherosclerotic plaques. By directly inhibiting the catalytic activity of these MMPs, H_2_S prevents excessive ECM degradation, maintaining the structural integrity of the fibrous cap. Additionally, the inhibition of CD147 by H_2_S leads to reduced MMP induction, further contributing to the stabilization of plaques. H_2_S also reduces the infiltration and activation of inflammatory cells, such as macrophages, within the plaque. Macrophages are significant sources of MMPs and other proteolytic enzymes contributing to plaque instability. By diminishing the recruitment of these cells through reduced expression of adhesion molecules like ICAM-1 and VCAM-1, H_2_S lowers the local production of MMPs and other inflammatory mediators. This reduction in inflammatory cell activity stabilizes the plaque and promotes a more quiescent plaque phenotype, less prone to rupture. The combined effects of H_2_S on MMP activity, CD147 expression, and inflammatory responses highlight its multifaceted role in enhancing plaque stability and reducing the risk of acute cardiovascular events. Future research should focus on optimizing dosing strategies and delivery methods for H₂S-based therapies while also evaluating their long-term safety and efficacy in preventing atherosclerosis progression. Detailed mechanistic studies are needed to clarify how H₂S modulates MMPs and CD147 at the molecular level. Additionally, exploring the role of H₂S in other cardiovascular and inflammatory conditions and identifying biomarkers for therapeutic response could enhance the clinical application of H₂S therapies. Personalized medicine approaches should also be considered to tailor treatments based on individual patient profiles.

## Figures and Tables

**Figure 1 biomedicines-12-01951-f001:**
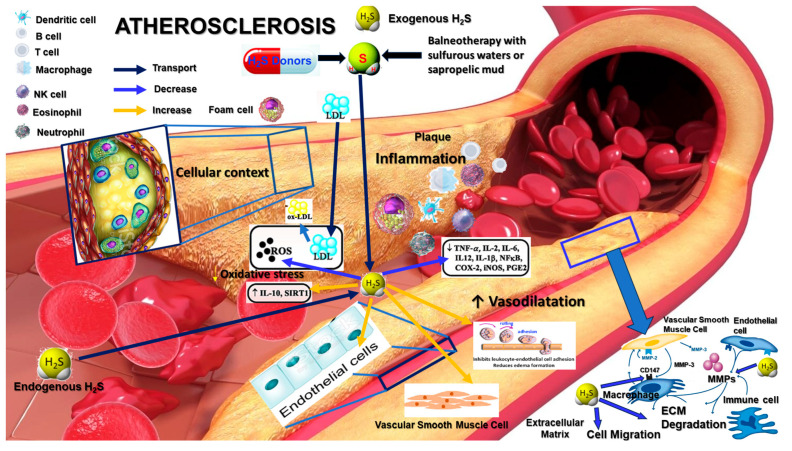
H_2_S pathways to modulate MMP and CD147 activity in atherosclerosis pathological context. H₂S reduces foam cell formation by decreasing oxidized LDL (ox-LDL) uptake, mitigates oxidative stress by scavenging reactive oxygen species (ROS) and upregulating protective enzymes like IL-10 and SIRT1, and controls inflammation by downregulating pro-inflammatory cytokines (e.g., TNF-α, IL-2, and IL-6) and NF-κB activity. The figure also demonstrates H₂S’s inhibition of MMP activity and CD147 expression, preserving extracellular matrix (ECM) integrity and stabilizing atherosclerotic plaques while promoting vasodilation and improved endothelial function.

**Table 1 biomedicines-12-01951-t001:** Different mechanisms by which hydrogen sulfide (H₂S) modulates matrix metalloproteinases (MMPs) and CD147/EMMPRIN.

Mechanism	Effect on MMPs	Effect on CD147/EMMPRIN	Outcome on Atherosclerosis	Ref. No.
**Direct Inhibition of MMP Activity**	H₂S directly interacts with the catalytic zinc ion in MMPs, inhibiting their proteolytic activity.	Not directly involved.	Stabilizes plaques by preventing excessive ECM degradation.	[54]
**Regulation of Oxidative Stress**	H₂S reduces oxidative stress by scavenging reactive oxygen species (ROS), preventing ROS-mediated activation of MMPs.	Reduces oxidative stress, indirectly decreasing CD147 expression and activity.	Reduces inflammation and ECM degradation, enhancing plaque stability.	[5,37,38,53]
**Modulation of NF-κB Signaling Pathway**	H₂S inhibits NF-κB activation, leading to decreased transcription of MMP genes such as MMP-9.	Inhibits NF-κB activation, reducing CD147 expression and MMP induction.	Lowers inflammation, decreases ECM degradation, and stabilizes plaques.	[37,58]
**S-Sulfhydration (Post-Translational Modification)**	Potentially alters MMP activity by modifying cysteine residues.	Modifies CD147, potentially affecting its interaction with MMPs and inflammatory signaling molecules.	Modulates protein function and signaling, contributing to reduced ECM degradation and inflammation.	[68,69]
**Inhibition of Smooth Muscle Cell (SMC) Migration and Proliferation**	Indirectly reduces MMP activity by inhibiting SMC migration and proliferation, key processes in plaque formation.	Not directly involved.	Limits plaque growth and promotes stability by maintaining ECM integrity.	[16,40,61,64,65]
**Interaction with Hypoxia-Inducible Factor 1-alpha (HIF-1α)**	Not directly involved.	H₂S modulates HIF-1α, reducing hypoxia-induced CD147 expression.	Decreases hypoxia-related inflammation and ECM degradation.	[28,70]

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
