# Peer review of "Hydrogen Sulfide Modulation of Matrix Metalloproteinases and CD147/EMMPRIN: Mechanistic Pathways and Impact on Atherosclerosis Progression"

_biomedicines, 2024, doi:10.3390/biomedicines12091951_

Round 1
Reviewer 1 Report
Comments and Suggestions for Authors
Congratulations. Exciting work. Minor comments in the file attached.

Author Response
Thank you for your comments provided for the manuscript titled "Hydrogen Sulfide Modulation of Matrix Metalloproteinases and CD147/EMMPRIN: Mechanistic Pathways and Impact on Atherosclerosis Progression," here is a detailed point-by-point response to address the concerns and suggestions:
Abstract:
- Comment: In the first three lines, the endothelium should be mentioned. (Lines 14-16)
- Response: We agree with the reviewer that the endothelium plays a crucial role in atherosclerosis. We have revised the abstract to mention the endothelium in the first three lines: "Atherosclerosis is a chronic inflammatory condition marked by endothelial dysfunction, lipid accumulation, inflammatory cell infiltration, and extracellular matrix (ECM) remodeling within arterial walls."
- Comment: On line 17, which type of cells have this glycoprotein?
- Response: The specific cell types expressing CD147/EMMPRIN have been clarified. The revised sentence now reads: "CD147/EMMPRIN, a glycoprotein expressed on endothelial cells, vascular smooth muscle cells (VSMCs), and immune cells, regulates MMP activity."
- Comment: On line 19, which processes? Be more specific.
- Response: We have specified the processes influenced by Hâ‚‚S in the revised text: "Hâ‚‚S influences various processes including oxidative stress mitigation, inflammation reduction, and vascular remodeling."
- Comment: On line 27, mention that the cellular proliferation is vascular.
- Response: The sentence has been updated to specify vascular proliferation: "Hâ‚‚S also downregulates CD147/EMMPRIN expression via transcriptional pathways, diminishing inflammatory responses and vascular cellular proliferation within plaques."
Manuscript Body:
- Comment: In line 167, the number 2 in Hâ‚‚S should be subindexed.
- Response: We have corrected the formatting issue - Hâ‚‚S 2 is subindexed throughout the manuscript.
- Comment: Consider moving the paragraph (lines 347-351) to line 307 for better organization and coherence.
- Response: We agree that this reorganization improves the manuscript's flow. The specified paragraph has been moved as suggested.
- Comment: Paragraphs in lines 335-340 and 406-413 contain some repetitive information. Consider revising to avoid redundancy.
- Response: We have revised the content of these paragraphs to eliminate redundancy, ensuring that each section contributes new and relevant information to the discussion.
- Comment: Consider deleting paragraph lines 369-372.
- Response: After reviewing the content, we concur with the suggestion. The paragraph in lines 369-372 has been removed.
- Comment: Can the glycosylation of CD147 be associated with glycotoxicity in T2DM?
- Response: We appreciate this insightful comment. We have added a paragraph to support this connection. ” . In Type 2 Diabetes Mellitus (T2DM), excessive glucose availability can lead to abnormal glycosylation patterns of CD147, potentially increasing its MMP-inducing activity. This abnormal glycosylation may exacerbate the inflammatory and degradative processes within the vascular walls, contributing to the accelerated progression of atherosclerosis observed in diabetic patients. Thus, the glycosylation of CD147 represents a plausible mechanistic link between glucotoxicity and vascular complications in T2DM, warranting further investigation into its role as a potential therapeutic target.”
- Comment: Could the lower right part of Figure 1 be made larger? Where the interaction of the cells on the plaque can be seen. It is tiny.
- Response: The figure has been revised to enlarge the lower right part, providing a clearer view of the interactions of cells on the plaque.
- Comment: On line 492, can the potential side effects be mentioned?
- Response: We have expanded the discussion in line 492 to include potential side effects associated with Hâ‚‚S modulation, emphasizing the importance of careful therapeutic application and potential risks: ” However, their rapid release can lead to short-lived effects and potential side effects, such as hypotension, respiratory distress, and neurological disturbances, necessitating careful dosing and monitoring. Overexpression or excessive administration of Hâ‚‚S could disrupt cellular homeostasis, leading to oxidative stress or interfering with mitochondrial function.”
We hope these revisions satisfactorily address the reviewer's concerns. We believe these changes have significantly strengthened the manuscript and appreciate the constructive feedback provided.
Reviewer 2 Report
Comments and Suggestions for Authors
The manuscript focuses on the role of hydrogen sulfide (H2S) in modulating matrix metalloproteinases (MMPs) and CD147/EMMPRIN, which are significant in the progression of atherosclerosis. The manuscript provides a comprehensive understanding of the impact of H2S on atherosclerosis through a systematic review of relevant literature from PubMed, Scopus, and Web of Science databases. The detailed data analysis and explanation of the role of H2S in atherosclerosis add persuasiveness to the study. However, please address the following issues for revision:
- There is a lack of logical cohesion in some sections of the introduction. For instance, after discussing endothelial dysfunction and foam cells, there is an abrupt transition to vascular remodeling without clearly explaining the relationship between vascular remodeling, inflammatory factors, and plaque formation.
- Punctuation is missing in the sentence: "It is synthesized primarily through the enzymatic activities of cystathionine γ-lyase (CSE), cystathionine β-synthase (CBS), and 3-mercaptopyruvate sulfurtransferase (3-MST) (37)These three enzymatic pathways are tightly regulated to ensure proper H2S levels in the body." Attention to punctuation throughout the manuscript is crucial for maintaining the clarity and professionalism of the writing.
- The expression "H2S stabilizes plaques by modulating MMP activity, crucial enzymes involved in ECM degradation," is problematic and unclear. It requires revision for improved clarity.
- Strengthen the logical flow in the introduction section by clearly linking vascular remodeling to inflammation and plaque formation.
- Correct the punctuation error in the sentence .
- Clarify the statement regarding H2S modulation of MMP activity to enhance understanding.
Author Response
The manuscript focuses on the role of hydrogen sulfide (H2S) in modulating matrix metalloproteinases (MMPs) and CD147/EMMPRIN, which are significant in the progression of atherosclerosis. The manuscript provides a comprehensive understanding of the impact of H2S on atherosclerosis through a systematic review of relevant literature from PubMed, Scopus, and Web of Science databases. The detailed data analysis and explanation of the role of H2S in atherosclerosis add persuasiveness to the study.
- There is a lack of logical cohesion in some sections of the introduction. For instance, after discussing endothelial dysfunction and foam cells, there is an abrupt transition to vascular remodeling without clearly explaining the relationship between vascular remodeling, inflammatory factors, and plaque formation.
Response: We acknowledge the reviewer's observation and have revised the introduction to improve the logical flow. We have added transitional sentences to clearly link endothelial dysfunction, foam cell formation, and their subsequent role in vascular remodeling.
- Punctuation is missing in the sentence: "It is synthesized primarily through the enzymatic activities of cystathionine γ-lyase (CSE), cystathionine β-synthase (CBS), and 3-mercaptopyruvate sulfurtransferase (3-MST) (37)These three enzymatic pathways are tightly regulated to ensure proper H2S levels in the body." Attention to punctuation throughout the manuscript is crucial for maintaining the clarity and professionalism of the writing.
Response: Thank you, We have corrected the punctuation in the mentioned sentence. A thorough check for punctuation errors has been conducted throughout the manuscript.
- The expression "H2S stabilizes plaques by modulating MMP activity, crucial enzymes involved in ECM degradation," is problematic and unclear. It requires revision for improved clarity.
Response: We have revised this statement for clarity. The updated sentence now reads: "Hâ‚‚S stabilizes atherosclerotic plaques by inhibiting the activity of matrix metalloproteinases (MMPs), which are critical enzymes involved in the degradation of the extracellular matrix (ECM)." This change clarifies the role of MMPs in ECM degradation and how Hâ‚‚S modulates their activity to stabilize plaques.
Comments on the Quality of English Language
- Strengthen the logical flow in the introduction section by clearly linking vascular remodeling to inflammation and plaque formation.
- Correct the punctuation error in the sentence .
- Clarify the statement regarding H2S modulation of MMP activity to enhance understanding.
- Response: In addition to the specific revisions mentioned above, we have undertaken a comprehensive review of the manuscript to enhance the overall quality of the English language.
We are grateful for the constructive feedback and hope that these changes meet the reviewers' expectations.
Reviewer 3 Report
Comments and Suggestions for Authors
see attached comments.

Author Response
To improve the clarity and visual attractiveness of your review, consider using pertinent figures and tables. These visual aids can efficiently summarise vital information, demonstrate complicated topics, and make the text more interesting to readers. Incorporating these components will not only improve the overall presentation, but will also attract and hold the attention of a larger audience.
Response: We appreciate the reviewer's suggestion to enhance the manuscript's visual appeal. We have incorporated a table that compares the different mechanisms by which hydrogen sulfide (Hâ‚‚S) modulates matrix metalloproteinases (MMPs) and CD147/EMMPRIN, providing a concise summary.
I found the methodology section to be somewhat unclear and perhaps misplaced in the context of this manuscript. Since this is a comprehensive review rather than a systematic review, the inclusion of a detailed methodology section is not typically required. Comprehensive reviews generally synthesize existing knowledge and provide expert insights without the need for a structured methodological approach. I suggest removing or revising this section to better align with the purpose and structure of a comprehensive review, thereby enhancing the readability and focus of your manuscript.
Response: We agree with the reviewer's observation that a detailed methodology section is not typically required in a comprehensive review. In response, we have removed the methodology section from the manuscript. To ensure that readers understand the scope and sources of the review, we have instead provided a brief introductory paragraph in the introduction that explains the criteria for selecting the literature reviewed and the overall approach taken.
To streamline the organization of the manuscript, I recommend removing the "results" subheading, as this format is not typical for a narrative review. Instead, consider summarizing the key findings of your review in visual formats such as charts, graphs, or tables. These visual presentations can provide a clear and concise overview of the outcomes, making it easier for readers to grasp the information at a glance.
Response: We acknowledge the reviewer's suggestion regarding the manuscript's structure. We have removed the "Results" subheading to better align with the format of a narrative review and incorporated a summary within the introduction.
Reviewer 4 Report
Comments and Suggestions for Authors
Reviewing the review manuscript entitled, “Hydrogen Sulfide Modulation of Matrix Metalloproteinases and CD147/EMMPRIN: Mechanistic Pathways and Impact on Atherosclerosis Progression” by Munteanu C et al., this is an article focusing on H2S functions on atherosclerosis progression. H2S's function in arteriosclerosis is an important, and this review may be of great value. The authors need to address my concerns below.
The authors should make the article easier to read by using subtitles such as Atherosclerosis, Vascular remodeling, MMP and CD147, H2S, etc. The article is long and difficult to read in the introduction section.
The authors should make it clear to readers that H2S, originally known as a toxic gas, has organ-protecting effects in the body in the introduction section. At first glance, it seems that the organ-protecting effect of endogenous H2S is the only thing being highlighted. If the H2S level in the body is higher or lower than normal, the body not function normally.
In 4.1. Role of Hydrogen Sulfide in Atherosclerosis section, the authors should add a figure that shows endogenous H2S production process using CSE, CBS and 3-MST. In addition, Figure 1 has "Exogenious H2S." By what route is this gas transported into the body?
In Figure 1, it is hard to distinguish between transport and decrease. The authors should modify it.
As an H2S donor, the authors should attach a table summarizing the clinical trials currently underway.
Comments on the Quality of English LanguageMinor editing is required.
Author Response
Reviewer Comment: The authors should make the article easier to read by using subtitles such as Atherosclerosis, Vascular Remodeling, MMP and CD147, Hâ‚‚S, etc. The article is long and difficult to read in the introduction section.
Response: We appreciate the reviewer's suggestion to improve the manuscript's readability. In response, we have changed the manuscript structure and organized the content, making it easier for readers to follow the narrative and understand the distinct concepts discussed.
Reviewer Comment: The authors should make it clear to readers that Hâ‚‚S, originally known as a toxic gas, has organ-protecting effects in the body in the introduction section. At first glance, it seems that the organ-protecting effect of endogenous Hâ‚‚S is the only thing being highlighted. If the Hâ‚‚S level in the body is higher or lower than normal, the body may not function normally.
Response: We acknowledge the importance of clarifying the dual nature of Hâ‚‚S as both a toxic gas and an endogenous signaling molecule with organ-protecting effects. We have revised the introduction to clearly state that Hâ‚‚S, while historically recognized as a toxic gas, is also endogenously produced in the body where it plays protective roles. We have also discussed the balance of Hâ‚‚S levels, emphasizing that excessive and insufficient Hâ‚‚S can lead to pathological conditions.
Reviewer Comment: In the “Role of Hydrogen Sulfide in Atherosclerosis” section, the authors should add a figure that shows the endogenous Hâ‚‚S production process using CSE, CBS, and 3-MST.
Response: We agree that a visual representation of the endogenous Hâ‚‚S production process would enhance the reader's understanding. However, such a figure can be easily found in so many articles about Hâ‚‚S and will be redundant.
Reviewer Comment: In addition, Figure 1 has "Exogenous Hâ‚‚S." By what route is this gas transported into the body?
Response: Thank you for pointing out the need for clarification. We have revised Figure 1 to include details on the routes of exogenous Hâ‚‚S administration. Specifically, we have indicated that exogenous Hâ‚‚S can be administered through inhalation/ balneotherapy with sulfurous mineral waters or mud, or via Hâ‚‚S-releasing compounds (Hâ‚‚S donors).
Reviewer Comment: In Figure 1, it is hard to distinguish between transport and decrease. The authors should modify it.
Response: We acknowledge the need for clearer distinctions in Figure 1. We have modified the figure by using different color schemes and arrow styles to distinguish between transport and decrease. The legend has been modified accordingly.
Reviewer Comment: As an Hâ‚‚S donor, the authors should attach a table summarizing the clinical trials currently underway.
Response: We appreciate the reviewer's suggestion to provide an overview of ongoing clinical trials involving Hâ‚‚S donors. In response, we must mention that previously published articles covered this matter.
Reviewer 5 Report
Comments and Suggestions for Authors
Atherosclerosis, a chronic inflammatory condition characterized by lipid accumulation, inflammatory cell infiltration, and extracellular matrix (ECM) remodeling within arterial walls. Hydrogen Sulfide (Hâ‚‚S) acts as a significant modulator of matrix metalloproteinases (MMPs) and CD147/EMMPRIN, influencing atherosclerosis progression. Hâ‚‚S modulates MMP expression and activity through transcriptional regulation and post-translational modifications, reducing oxidative stress and inflammation. Hâ‚‚S shows promise as a therapeutic agent in stabilizing atherosclerotic plaques and mitigating inflammation, with further research needed to explore its clinical applications. Specific comments:
1. The introduction provides a comprehensive overview of atherosclerosis. However, it could benefit from a more concise summary of the key points to improve readability.
2. The paper discusses the modulation of MMPs and CD147/EMMPRIN by Hâ‚‚S. Could the authors provide more detailed mechanistic pathways or diagrams to illustrate these interactions?
3. The literature review is thorough, but it would be helpful to include a table summarizing the key findings of the studies reviewed.
4. The methodology section is well-detailed. However, it would be beneficial to include a flowchart of the literature search and selection process.
5. The paper includes a figure. It would be helpful to ensure that the figure is clearly labeled and referenced in the text.
6. The discussion section is insightful. However, it could benefit from a more in-depth analysis of the potential clinical applications of Hâ‚‚S-based therapies.
7. The conclusion summarizes the key findings well. Could the authors provide more specific recommendations for future research based on their findings?
Author Response
Atherosclerosis, a chronic inflammatory condition characterized by lipid accumulation, inflammatory cell infiltration, and extracellular matrix (ECM) remodeling within arterial walls. Hydrogen Sulfide (Hâ‚‚S) acts as a significant modulator of matrix metalloproteinases (MMPs) and CD147/EMMPRIN, influencing atherosclerosis progression. Hâ‚‚S modulates MMP expression and activity through transcriptional regulation and post-translational modifications, reducing oxidative stress and inflammation. Hâ‚‚S shows promise as a therapeutic agent in stabilizing atherosclerotic plaques and mitigating inflammation, with further research needed to explore its clinical applications.
We sincerely thank you for the constructive feedback provided by the reviewer.
Specific comments:
- The introduction provides a comprehensive overview of atherosclerosis. However, it could benefit from a more concise summary of the key points to improve readability.
Response: We have streamlined the introduction to provide a more concise summary of the key points, improving readability without compromising the essential background information on atherosclerosis and the role of Hâ‚‚S.
- The paper discusses the modulation of MMPs and CD147/EMMPRIN by Hâ‚‚S. Could the authors provide more detailed mechanistic pathways or diagrams to illustrate these interactions?
Response: A new figure has been added to visually represent the pathways through which Hâ‚‚S modulates MMP expression and CD147/EMMPRIN activity.
- The literature review is thorough, but it would be helpful to include a table summarizing the key findings of the studies reviewed.
Response: We have included a table summarizing the key findings from the reviewed studies, offering a clear and concise overview of the evidence discussed in the manuscript.
- The methodology section is well-detailed. However, it would be beneficial to include a flowchart of the literature search and selection process.
Response: We appreciate the suggestion. In response to a previous reviewer's feedback, we were advised that the detailed methodology section might be more appropriate for a systematic review rather than a comprehensive review like ours. As a result, we decided to streamline the methodology section to focus more on synthesizing existing knowledge and providing expert insights, aligning with a comprehensive review's scope. Given this context and the nature of our review, we have chosen not to include a flowchart for the literature search and selection process, as our focus is on integrating and discussing the current literature rather than detailing the search methodology.
- The paper includes a figure. It would be helpful to ensure that the figure is clearly labeled and referenced in the text.
Response: The existing figure has been carefully reviewed to ensure it is clearly labeled and appropriately referenced within the text.
- The discussion section is insightful. However, it could benefit from a more in-depth analysis of the potential clinical applications of Hâ‚‚S-based therapies.
Response: We acknowledge the concern regarding potential redundancy with previously published articles. However, our review uniquely synthesizes the latest research on the modulation of MMPs and CD147 by Hâ‚‚S specifically in the context of atherosclerosis, focusing on recent mechanistic insights and therapeutic implications. While some foundational concepts may overlap with existing literature, our review expands on these by incorporating the most current studies and offering a comprehensive perspective on the emerging therapeutic potential of Hâ‚‚S.
- The conclusion summarizes the key findings well. Could the authors provide more specific recommendations for future research based on their findings?
Response: We have added a paragraph providing recommendations aim to guide future research efforts in harnessing the therapeutic potential of Hâ‚‚S while addressing key gaps in current knowledge.
We hope these revisions and clarifications address the reviewer's concerns and improve our manuscript's overall quality and focus.